# Helsmoortel–Van der Aa Syndrome—Cardiothoracic and Ectodermal Manifestations in Two Patients as Further Support of a Previous Observation on Phenotypic Overlap with RASopathies

**DOI:** 10.3390/genes13122367

**Published:** 2022-12-15

**Authors:** Tímea Margit Szabó, István Balogh, Anikó Ujfalusi, Zsuzsanna Szűcs, László Madar, Katalin Koczok, Beáta Bessenyei, Ildikó Csürke, Katalin Szakszon

**Affiliations:** 1Institute of Paediatrics, Faculty of Medicine, University of Debrecen, 4032 Debrecen, Hungary; 2Division of Clinical Genetics, Department of Laboratory Medicine, Faculty of Medicine, University of Debrecen, 4032 Debrecen, Hungary; 3Jósa András Hospital, Szabolcs-Szatmár-Bereg County Hospitals and University Teaching Hospital, 4400 Nyíregyháza, Hungary

**Keywords:** Helsmoortel–Van der AA syndrome, *ADNP*, Noonan

## Abstract

The *ADNP*-gene-related neurodevelopmental disorder Helsmoortel–Van der Aa syndrome is a rare syndromic-intellectual disability—an autism spectrum disorder first described by Helsmoortel and Van der Aa in 2014. Recently, a large cohort including 78 patients and their detailed phenotypes were presented by Van Dijck et al., 2019, who reported developmental delay, speech delay and autism spectrum disorder as nearly constant findings with or without variable cardiological, gastroenterological, urogenital, endocrine and neurological manifestations. Among cardiac malformations, atrial septal defect, patent ductus arteriosus, patent foramen ovale and mitral valve prolapse were the most common findings, but other unspecified defects, such as mild pulmonary valve stenosis, were also described. We present two patients with pathogenic *ADNP* variants and unusual cardiothoracic manifestations—Bland–White–Garland syndrome, pectus carinatum superiorly along the costochondral junctions and pectus excavatum inferiorly in one patient, and Kawasaki syndrome with pericardiac effusion, coronary artery dilatation and aneurysm in the other—who were successfully treated with intravenous immunoglobulin, corticosteroid and aspirin. Both patients had ectodermal and/or skeletal features overlapping those seen in RASopathies, supporting the observations of Alkhunaizi et al. 2018. on the clinical overlap between Helsmoortel–Van der Aa syndrome and Noonan syndrome. We observed a morphological overlap with the Noonan-like disorder with anagen hair in our patients.

## 1. Introduction

Pathogenic variants of the *ADNP* gene (MIM 611386) cause Helsmoortel–Van der Aa syndrome (MIM 615873), an autosomal dominant neurodevelopmental disorder characterised by intellectual disability, motor delay (ID/DD), autism spectrum disorder (ASD), hypotonia, speech delay and variable other features, such as congenital heart defects, distinctive facial appearance, urogenital anomalies, ophthalmological, gastrointestinal and behavioural problems [1]. The syndrome was first described by Helsmoortel and Van der Aa et al. in 2014. [2], two years after O’Roak et al. reported a frameshift indel variant in the *ADNP* gene as one of the “top de novo ASD risk contributing mutations” in a large study of 189 autism trios investigated by whole-exome sequencing [3]. *ADNP* is an evolutionarily conserved gene that encodes the activity-dependent neuroprotective protein (ADNP). It was discovered in the laboratory of Pr. Illana Gozes; its amino acid sequence was established by Bassan et al. [4,5]. Structurally, it contains nine zinc fingers, a proline-rich region, a nuclear bipartite localisation signal and a homeobox domain profile consistent with its role as a transcription factor [6].

The ADNP protein is highly expressed in the brain and is involved in regulating gene expression and chromatin remodelling via its interaction with members of the switching/sucrose non-fermenting (SWI/SNF) remodelling complex (BAF complex) [7,8]. Genes that are subject to transcriptional repression control cell-fate decisions [9,10]. 

ADNP may be linked to cognitive ability [7,11,12]; it plays an important role in neurodevelopment [7], and exhibits a neuroprotective function attributed to its octapeptide NAPVSPIQ (Asn-Ala-Pro-Val-Ser-Ile-Pro-Ala) domain called NAP [5,13] that has recently been acknowledged as a potential therapeutic target [13,14]. 

Since the initial description of the first ten patients with Helsmoortel–Van der Aa syndrome [2], there have been an increasing number of reports on further patients with this condition [15,16,17,18,19]. *ADNP* has been proposed as one of the most frequent ASD-associated genes as it is estimated to be mutated in about 0.17% of ASD cases [2]. The largest cohort of 78 patients was published by Van Dijck et al., 2019, providing detailed phenotypic and genotypic descriptions and frequency of symptoms, and supporting early clinical recognition, diagnosis and management of affected individuals [20]. 

In this report, we present two patients with pathogenic *ADNP* variants and unusual cardiothoracic and ectodermal findings that—to our knowledge—have not yet been described in individuals with Helsmoortel–Van der Aa syndrome. Our report is the second until now highlighting clinical overlap with Noonan syndrome, supporting the observations of Alkhunaizi et al. [21].

## 2. Materials and Methods

Genetic testing was performed after obtaining written informed consent from parents. Genetic tests were performed as part of routine diagnostic genetic testing, according to the Declaration of Helsinki. Informed consent for the use of medical data and photo documentation was obtained from the parents of patients.

Genomic DNA was isolated from peripheral blood leukocytes using the QIAamp Blood Mini kit (Qiagen GmbH, Hilden, Germany).

Copy number changes were investigated using a Affymetrix CytoScan 750K and Affymetrix Chromosome Analysis Suite (ChAS) v2.0 Software (Affymetrix, Thermo Fisher Scientific, Waltham, MA, USA).

Clinical Exome Sequencing (for Patient 1) was performed on the Illumina MiSeq sequencer system in 2 × 150 cycle paired-end mode. A TruSight One Sequencing Panel kit (Illumina, San Diego, CA, USA) was used for library preparation. Whole-exome sequencing (WES) (for Patient 2) was performed on the Illumina NextSeq 500 sequencer system in 2 × 150 cycle paired-end mode. A Nextera DNA Exome kit (Illumina) was used for library preparation. For both patients, Sanger confirmation was performed using the BigDye Terminator v3.1 Cycle Sequencing kit (Applied Biosystems, Foster City, CA, USA) according to the manufacturer’s protocol. For Patient 1, the following primers were used: *ADNP*-e4-5′-TGCAGACAAAAAGACTTTGGAA-3′ forward primer and 5′-TGGTCTTCGATGACATGCTG-3′ reverse primer. For Patient 2, *ADNP*-e6 5′-CAACATGACCGCCTCAACTA-3′ forward primer and e6 5′-CACGCCAGGCTTGTACTTTT-3′ reverse primers were used. Exome sequencing raw data were aligned to the hg19 reference genome using the NextGene software (SoftGenetics, State College, PA, USA). Variants were filtered based on severity and frequency against public variant databases, including ClinVar, HGMD Professional, gnomAD and in-house exome frequency databases. The variant list was also uploaded to the Franklin Analysis Platform (Genoox) for variant classification, and HPO terms were used to help with variant prioritisation.

Variants in the *ADNP* gene were designated based on the reference sequence NM_181442.1 for the coding sequence of the gene and NP_852107.1.

Anthropometric data were matched against Hungarian child growth standards [22,23].

## 3. Patients and Results

Patient 1 is a female patient, born at term as the second child of healthy, non-consanguineous Caucasian parents per vias naturales with normal birth parameters; weight: 3150 g (50th pc); length: 50 cm (50th pc); head circumference: 33 cm (25th−50th pc). A cardiac murmur noted at 2 weeks of age prompted echocardiography upon which grade II mitral insufficiency and an 11 mm, large, haemodynamically significant atrial septal defect were seen. At 6 months old, she underwent atrial septal defect closure using the patch-repair technique. An intercurrent respiratory syncytial virus (RSV) infection caused life-threatening pneumonia requiring mechanical ventilation and resuscitation. Cardiological follow-up detected a coronary fistula; due to this, coronary artery catheterisation and coronary CT angiogram were performed and revealed Bland–White–Garland syndrome: an anomalous, stenotic left coronary artery originating from the truncus pulmonalis, being retrograde-filled from the dilated right coronary artery via anastomoses. To correct the origin of the left coronary artery she underwent another operation at the age of 5.5 years. 

Developmental milestones were delayed: sitting was achieved at 14 months, walking independently was achieved at 2.5 years of age, and the first spoken words were reported by the parents at 5 years of age. She has attention deficit, apparent insensitivity to pain, sleeping and behavioural problems and occasional aggression or tantrums. Her overall intelligence falls into the range of moderate intellectual disability. 

Referral to a clinical geneticist occurred at 7 years of age; at this time, weight was 17 kg (1.5 kg < 3rd pc), height was 112 cm (10–25th pc), and head circumference was 47.4 cm (1 cm < 3rd pc). Extreme shyness, anxiety, avoidance of eye contact, fairly good verbal comprehension, spontaneous but often meaningless, unintelligible speech, articulation problems and refusal to engage in conversations or answer questions were remarkable, suggesting the presence of an autism spectrum disorder. Morphologically, she had a normal skull shape, high frontal and posterior hairline, thin, straight, sparse hair and hypotrichosis of the eyebrows and cutis marmorata. She had downslanted and long palpebral fissures, mild ptosis, everted and somewhat hanging lower eyelids, strabism and a peculiar upward gaze resembling Individual 39 from the cohort of Van Dijck. et al. [1]. She had a narrow and high nasal root, long columella, mildly hypoplastic alae nasi, an open-mouth appearance with a thin upper lip and dental crowding. Primary and permanent dentition were both early, at 2.5 months and at 4.5–5 years, respectively. The ears were normally formed, with evenly thick helices. On the chest, there was pectus carinatum superiorly along the costochondral junctions and pectus excavatum inferiorly, evident at the first clinical genetic evaluation, but not in infancy. (Figure 1A,B).

The morphology of the eyes resembled Kabuki syndrome, but the *KMT2D* gene (MIM 602113) direct Sanger sequencing (performed in an external laboratory: Clinical Genomics Laboratory, Academic Hospital, Maastricht, The Netherlands) had negative results. Array CGH did not reveal pathological CNVs, also ruling out DiGeorge syndrome (MIM 188400). Personal communication with Prof. Raoul Hennekam (University Medical Centers, Amsterdam, The Netherlands) raised the possibility that the co-occurrence of cardiac septal and valvular defect, thin hair, thoracic malformation, intellectual disability and downslanted palpebral fissures may, in fact, be related to an underlying RASopathy. Therefore, next-generation sequencing of the coding regions of 13 genes—*BRAF* (MIM 164757), *CBL* (MIM 165360), *HRAS* (190020), *KAT6B* (MIM 605880), *KRAS* (MIM 190070), *MAP2K1* (MIM 176872), *MAP2K2* (MIM 601263), *NF1* (MIM 613113), *NRAS* (MIM 164790), *PTPN11* (MIM 176876), *RAF1* (MIM 164760), *SOS1* (MIM 182530) and *SHOC2* (MIM 602775)—was performed by Gendia, Belgium, and did not reveal any pathogenic variants. The time of the presentation of the patient coincided with the first publication on Helsmoortel–Van der Aa syndrome, which explains our serial unsuccess to diagnose her. When the patient was 12 years old, clinical exome sequencing became available to us, allowing the analysis of 4813 morbid OMIM genes, and a de novo pathogenic frameshift mutation in the *ADNP* gene c.539_542delTTAG, p.(Val180Glyfs*17) was identified in a heterozygous state, as has been described in the literature [1,24]. ClinVar ID: 373314. Interestingly, this patient shows striking facial similarity to Patient 39 in the cohort of Van Dijck et al.; the facial gestalt and upward gaze are almost identical, just like their genotypes. The hair of Patient 39 in the Van Dijck et al. study is apparently also thin, but not sparse. [1]

Patient 2 is the first child of healthy, non-consanguineous Caucasian parents, born in the 37th week of gestation by C-section due to maternal high-grade myopia, with the following birth parameters: weight: 2480 g (10th−25th pc); length: 44 cm (1.5 cm < 3rd pc); head circumference: 33.5 cm (50th pc). A heart murmur was noted soon after birth; cardiac ultrasound revealed an atrial septal defect and pulmonary valve stenosis. At 2 weeks of age, she had a urinary tract infection, severe jaundice and anaemia requiring a transfusion. A short lingual frenulum necessitated surgical intervention.

At the time of the first clinical genetic examination, she was 10 months old. She had a high-pitched, hypernasal voice, round face, bilateral ptosis, mild epicanthal folds, horizontal palpebral fissures, vertically broad and sparse eyebrows, mildly concave nasal ridge, long philtrum, small mouth with thin upper lip and maxillary prognathism. Her hair was very thin, sparse and straight, laterally short and somewhat longer on the top of the head. According to parental reports, the texture and thickness of her hair normalised by 5 years of age. On her hands, she had 5th finger clinodactyly, prominent distal interphalangeal joint of the left 2nd finger and bilateral 2/3 toes syndactyly (Figure 1C,D). Her nails were, and still are, thin and brittle. Her labia minora were hypoplastic. At 3 years old, her somatic parameters were normal with a weight of 14.5 kg (50th−75th pc); length of 90.5 cm (10th−25th pc); head circumference of 49.5 cm (50th pc).

Her motor development was delayed, she was able to sit alone at the age of 9 months, commando-crawl at 13 months, climb at 18 months and walk unassisted at 24 months. Intellectual and speech development were also delayed and she began to say her first words at 3 years old. She has a mild intellectual deficit with hyperactivity–attention deficit disorder, sleep problems and mild autistic traits that present with stereotypical movements of the hands. Temper tantrums are rare. A high pain threshold, gastroesophageal reflux, difficulty with chewing solid foods and frequent vomiting were reported.

At the age of 18 months, she developed atypical Kawasaki syndrome with elevated C-reactive protein (155 mg/L; Ref: <4 mg/L), and as complications, right coronary artery dilatation (2.2 mm diameter), left coronary artery aneurysm (2.7 mm diameter) and pericardiac effusion developed. Cultures did not confirm a bacterial origin and SARS-CoV-2, which resulted in multisystemic inflammatory syndrome in children (MISC), was not known in the year 2017. She was treated according to the corresponding guideline of the American Heart Association with intravenous immunoglobulin, corticosteroids and acetyl-salicylate and recovered without further complications [25,26].

To clarify the cause of developmental delay, array CGH was performed and a paternal 407 kb duplication of the Xp22.2(13495629_13902747) was found and considered benign. DiGeorge (MIM 188400) and Smith–Magenis syndrome (MIM 182290), resulting from pathological CNVs, were therefore ruled out—the latter was suspected based on self-clutching and stereotypic movements of the hands. Using whole-exome sequencing, a heterozygous, de novo nonsense variant c.2213C>A, p.(Ser738*) in the *ADNP* gene was identified and classified as pathogenic [1,24]. ClinVar ID 984838. Our Patient 2 has the same mutation (p.Ser738*), as Patient 71 in the cohort of Van Dijck et al. [1]. Individual phenotypic description for that patient is not available. We did not find a facial photograph of an individual with the p.Ser738* variant either in the Helsmoortel et al., 2014. paper [2].

A clinical table containing the occurrence of Helsmoortel–Van der Aa syndrome-related features in our patients compared to previously reported ones was prepared, based on the table of Van Dijck et al., 2019 [1]. (Table 1).

## 4. Discussion

The *ADNP*-gene-related Helsmoortel–Van der Aa syndrome is a rare but increasingly often reported form of syndromic intellectual disability and autism. Intellectual deficit, motor developmental delay, hypotonia, speech delay, behavioural problems, an often recognisable facial gestalt and variable other symptoms, such as cardiovascular, ocular, neurological, gastrointestinal, dental and urogenital anomalies characterise the clinical picture [1]. Accumulated evidence suggests a dominant-negative mutational mechanism; most mutations being nonsense or frameshift variants resulting in a premature stop codon and truncated protein, escaping nonsense-mediated mRNA-decay [1]. To date, about a hundred individuals are confirmed to be affected [1,10,27,28], but the true prevalence is presumably higher [2].

Understanding ADNP’s mediating role in the formation of the (SWI/SNF) remodelling (BAF) complex, it is not surprising that the Helsmoortel–Van der Aa syndrome shares phenotypic similarities with BAFopathies [8,9,10], such as Coffin–Siris syndrome (MIM 135900), with the following common features: developmental delay; gastrointestinal problems; visual impairment; hypotonia; central nervous system abnormalities; a small or hypoplastic distal phalanx of the 5th fingers; facial similarity [1,17]. Resemblance to Kleefstra (MIM 610253), Smith–Magenis (MIM 182290), Prader–Willi (MIM 176270), Rett syndrome (MIM 312750) and even Angelman syndrome (MIM 105830) has been described with special regards to the behavioural phenotype [29]. 

There is little clinical evidence on an overlap with RASopathies—Alkhunaizi et al. reported an ADNP-patient with features of Noonan syndrome, including ptosis, flat supraorbital ridges, webbed neck with low nuchal hairline, widely-spaced nipples and pectus carinatum [21,29].

We presented two patients with pathogenic *ADNP* variants: one harbouring a p.(Val180Glyfs*17) frameshift and the other a p.(Ser738Ter) nonsense variant resulting in a premature termination codon. Both patients presented with the core features of *ADNP* syndrome, including developmental delay, intellectual disability, congenital heart defects, ophthalmological problems, autism spectrum disorder/attention deficit–hyperactivity disorder, decreased sensitivity to pain and gastrointestinal problems (gastroesophageal reflux in Patient 2). Notably, our patients share features of RASopathies, such as thin, sparse, straight—but not easily pluckable—hair, vertically broad and sparse eyebrows, ptosis, antimongoloid slant of the palpebral fissures, cardiac defects and thoracic wall deformity. The hair remained slow-growing in Patient 1, but its thickness normalised in Patient 2 with advancing age. Other ectodermal features were the hypotrichosis of the eyebrows in both patients, thin, brittle nails in Patient 2 and early primary dentition in Patient 1, but the latter is not a feature seen in RASopathies; it is only characteristic of Helsmoortel–Van der Aa syndrome [24]. Improvement of hair growth has been described in the RASopathy cardi–facio–cutaneous syndrome (MIM 115150) [30] due to the natural course of improvement of skin dryness and hyperkeratosis—we did not detect the latter two cutaneous symptoms in our patients. Patient 1 has a strong facial resemblance to individuals with *SHOC2*-related Noonan-like disorder with loose anagen hair (MIM 607721) reported by Capalbo et al., 2012, and Hannig et al., 2013 [31,32]. Over-representation of mitral valve dysplasia and septal defects in Noonan-like syndrome with loose anagen hair in comparison with classic Noonan syndrome was also described by Hannig et al. [32] (Patient 1 in our study had atrial septal defect and grade II mitral valve prolapse). Pectus carinatum superiorly and pectus excavatum inferiorly in Patient 1 is frequently described in Noonan syndrome [30]. Although cardiac surgery through sternotomy was performed at 6 months of age in our Patient 1, the symmetrical prominence along the costochondral junctions of the chest and not along the sternum suggest a malformation rather than a deformation. Observations regarding the overgrowth of costal cartilage and undergrowth of ribs in patients with bilateral pectus carinatum and the progressive nature of pectus carinatum support our reasoning [33,34]. To our knowledge, the unique coronary artery anomaly Bland–Garland–White syndrome is first reported here in relation to *ADNP* syndrome. The pulmonary artery stenosis in Patient 2 is a more common cardiac anomaly in Noonan- as well as in Helsmoortel–Van der Aa syndrome. Short stature or macrocephaly were not seen in our patients. Kawasaki disease or coronary artery aneurysm have been scarcely reported in Noonan syndrome patients [35,36], and we have no knowledge on it ever being reported in Helsmoortel–Van der Aa syndrome. Kawasaki disease as a comorbidity in Patient 2 may, therefore, be just a single example. According to our observations, the clinical features of our *ADNP* patients—apart from their known underlying genetic condition defining their appearance and coexisting other health issues—show remarkable clinical overlap with the facial gestalt of Noonan-like disorder with loose anagen hair. The clinical features found among our *ADNP* patients which overlapped *SHOC2*-related Noonan-like syndrome with loose anagen hair based on the table of Capalbo et al., 2012. are presented in Table 2.

## 5. Conclusions

By reporting not-yet-described clinical features in two *ADNP* patients and highlighting the partial phenotypic overlap between *ADNP*-related Helsmoortel–Van der Aa syndrome and RASopathies, our aim was to provide further evidence supporting the initial observations of Alkhunaizi et al., 2018, to broaden the clinical spectrum of Helsmoortel–Van der Aa syndrome and draw attention to possible difficulties in the differential diagnosis. By applying precise phenotyping, Helsmoortel–Van der Aa syndrome may be clinically recognisable, but a “genotype-first” approach may be needed to establish the diagnosis of patients, which then may help patients and families obtain access to treatment options. In *ANDP* syndrome, promising results have been achieved with low-dose ketamine in a study of 10 patients, independent from their genotype [37].

## Figures and Tables

**Figure 1 genes-13-02367-f001:**
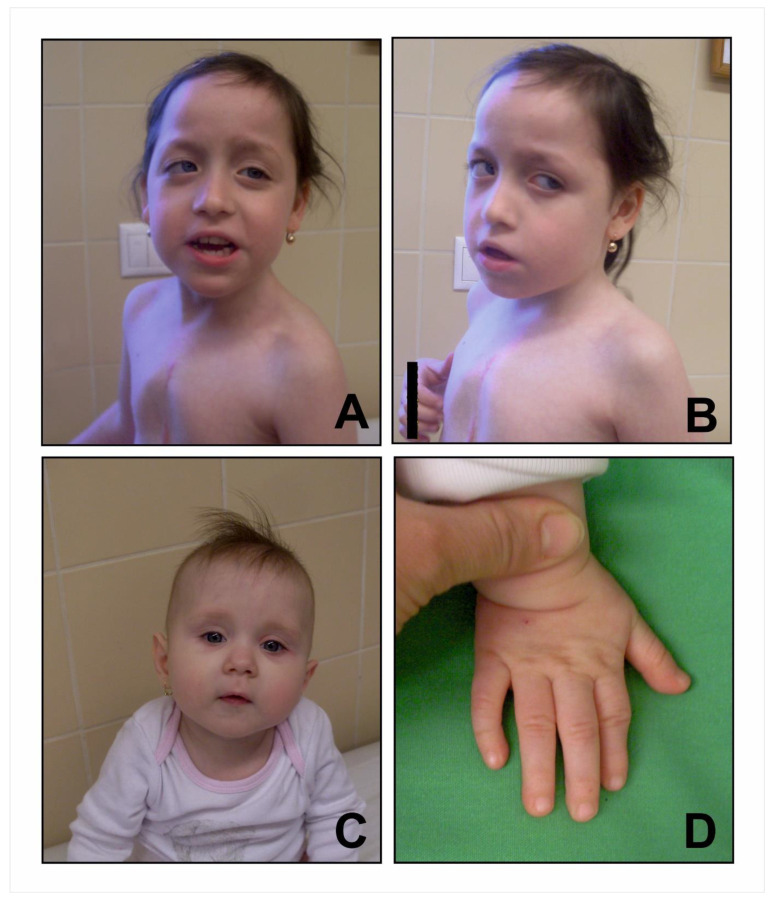
Photographs of patient with pathogenic *ADNP* variants. Note thin, straight, sparse hair; light skin colour; ptosis; high anterior and posterior hairline in both patients (**A**–**C**); long and antimongoloid palpebral fissures; strabismus; peculiar upward gaze; low-hanging, everted lower eyelids; an evenly thick helix bilaterally; superior pectus carinatum; inferior pectus excavatum and the scar of previous cardiac surgeries in Patient 1 (**A**,**B**). Round face; horizontal palpebral fissures; ptosis; long philtrum; small mouth; thin upper lip; apparent hypotonia (**C**); 5th finger clinodactyly and prominent distal interphalangeal joint of the 2nd finger (**D**) can be seen in Patient 2.

**Table 1 genes-13-02367-t001:** Comparison of patients’ symptoms to the results of Van Dijck et al., 2019.

Clinical Feature	Results of Van Dijck et al., 2019 (Sample N/Total N)	This Study P1	This Study P2
**General information**			
Age at examination	1–40 years	7 years	11 months
Gender (female:male)	34:44	female	female
Gestational age (weeks)	38.7	40	37
Age of father at time of birth (years)	32.1	31	28
Age of mother at time of birth (years)	29.8	28	25.5
**Mutation information**			
De novo *ADNP* mutation	97.1%	+	+
Nonsense mutation	56.4%	−	+ c2213C>A, p.(Ser738*)
Frameshift mutation	43.6%	+ c.539_542delTTAG, p.(Val180Glyfs*17)	−
**Growth**			
Short stature, <2 SD	23.2%	−	+ (prenatally)
**Neurodevelopmental features**			
Developmental delay/ID	100%	+	+
Mild ID	12.3%	−	+
Moderate ID	35.6%	+	−
Severe ID	52.1%	−	−
Motor delay			
Age at sitting independently, years (mean)	1.1	1.16	0.75
Walking independently	86.8%	+	+
Age at walking independently, years (mean)	2.5	2.5	2
Speech delay	98.6%	+	+
Age at first words, years (mean)	2.5	5	3
No speech	19.4%	−	−
Autism spectrum disorder	92.8%	+	+
Attention-deficit/hyperactivity disorder	43.9%	+	+
Loss of skills	20.3%	−	−
Bladder training delay	81.1%	+	+
**Feeding and gastrointestinal problems**			
Gastroesophageal reflux (disease)	58.5%	−	+
Constipation	49.3%	−	−
Oral movement problems	45.6%	−	−
Lack of satiations	41.5%	−	+
Problems swallowing liquids	32.2%	−	−
Frequent vomiting	29.5%	−	+
Aspiration difficulties	21.4%	−	−
Gastrostomy tube	12.7%	−	−
Obesity	70.5%	−	−
**Neurological problems and behaviour**			
Hypotonia	78.3%	−	+
Hypertonia	3.8%	−	−
Seizures	16.2%	−	−
Cerebral imaging-structural brain abnormalities	55.9%	NA	−
Wide ventricles	29.4%	NA	−
Corpus callosum underdevelopment	18.4%	NA	−
Cerebral atrophy	17.8%	NA	−
Delayed myelination	8.9%	NA	NA
White matter lesions	7.5%	NA	−
Cortical dysplasia	3.8%	NA	−
MRI brain abnormalities, unspecified	36.2%	NA	−
Behavioural problems	77.6%	+	+
Temper tantrums/aggression	83.3%	+	+
Obsessive–compulsive behaviour	64.0%	−	−
Mood disorder	56.3%	−	−
Self-injurious behaviour	20.0%	−	−
Insensitivity to pain	63.6%	+	+
Sensory processing disorder	66.7%	NA	NA
Sleep problems	65.2%	+	+
**Visual system**	73.6%	+	+
Strabism	49.2%	+	−
Central visual impairment	41.2%	−	−
Hypermetropia	40.3%	+	−
Ptosis	24.2%	+	+
Nystagmus	11.7%	−	−
Myopia	7.9%	−	−
Colobomata	5.6%	−	−
**Ear-Nose-Throat System**	32.1%	−	+
Narrow hearing canal	87.5%	−	−
Frequent otitis media	85.7%	−	−
Hearing tubes	73.3%	−	−
Hearing loss	11.7%	−	−
Obstructive sleep apnoea syndrome	6.6%	−	+
**Cardiovascular system**	37.7%	+	+
Atrial septal defect	15.9%	+	+
Patent ductus arteriosus	8.7%	−	+
Mitral valve prolapse	5.8%	NA	NA
Patent foramen ovale	5.8%	−	−
Ventricular septal defect	4.3%	−	−
Tetralogy of Fallot	1.4%	−	−
Cardiac defect, unspecified	8.7%	+ (BWG, MI)	+ (PS)
**Urogenital system**	28.0%	−	+
Cryptorchidism	34.3%		
Renal anomalies	12.5%	−	−
Small genitalia	5.4%	−	+
**Endocrine system**	24.5%	+	−
Early puberty	30.0%	−	−
Thyroid hormone problems	15.2%	NA	−
Growth hormone deficiency	10.9%	−	−
**Musculoskeletal system**	54.9%	+	−
Joint hypermobility	37.7%	−	−
Scoliosis	17.2%	−	−
Hip problems (hip dysplasia, Perthes’ disease, dislocated hips)	7.5%	−	−
Thorax abnormalities	22.2%	+	−
Pectus excavatum	14.8%	+	−
Pectus carinatum	5.6%	+	−
Narrow thorax	1.9%	−	−
Abnormal skull shape	13.9%	−	−
Plagiocephaly	8.3%	−	−
Trigonocephaly	2.8%	−	−
Brachycephaly	4.2%	−	−
**Hand and foot abnormalities**	62.3%	−	+
Finger abnormalities (prominent distal phalanges, prominent interphalangeal joints, polydactyly, interdigital webbing, 2–3 toe syndactyly, fifth finger clinodactyly, small fifth finger or absent distal phalanx of fifth finger, tapering fingers, brachydactyly, broad fingers, foetal fingertip pads)	46.3%	−	+
Single palmar crease	10.8%	−	−
Nail anomalies	25.0%	−	−
Sandal gap	19.6%	−	−
Toe abnormalities (broad halluces, 2–3 toe syndactyly, brachydactyly)	10.8%	−	+
**Other**			
Early teeth	71.1%	+	−
Frequent infections	50.7%	−	−
Widely spaced nipples	20.4%	−	−
Umbilical/inguinal hernia	8.5%	−	−

ID: intellectual disability; MRI: magnetic resonance imaging; BWG: Bland–White–Garland syndrome; MI: mitral insufficiency; PS: pulmonary stenosis; NA: not assessed/not available.

**Table 2 genes-13-02367-t002:** Comparison of findings in current patients to the patients of Capalbo et al., 2012.

Clinical Feature	Capalbo, 2012, P1	Capalbo, 2012, P2	This Study P1	This Study P2
Skin and hair				
Dark skin	+	+	− (light)	− (light)
Hyperkeratosis	−	−	−	−
Dermatitis/eczema	−	−	−	−
Sparse/absent scalp hair	+	+	+	+ (at young age; normalised)
Curly hair	−	−	−	−
Loose anagen hair	+	+	anagen	−
Head and neck				
Macrocephaly	+	+	− (micro)	−
Eyes				
Ptosis	−	+	+	+
Epicanthal folds	+	+	−	+ (mild)
Downslanted palp. fissures	+	−	+	−
Everted/low hanging lower eyelids	+	+ (mild)	+	−
Nose				
Depressed nasal bridge	−	−	−	−
Anteverted nostrils	−	−	−	−
Oral cavity				
High-arched palate	−	+	NA	NA
Ears				
Apparently low-set ears	+	−	−	−
Retroverted ears	+	−	−	−
Neck				
Broad, short, webbed	+	+	−	−
Chest				
Pectus excavatum	+	+	+, +carinatum	−
Cardiovascular				
VSD/ASD/PDA	+	−	+, + BWG, MI	+, PS
Cerebral abnormalities				
Dilated ventricles	−	−	NA	−
Intellectual disability	−	+	+	+
Tic	−	+	−	−
Growth and development				
Short stature	+	+	−	+ (prenatally)
Lymphatic dysplasia	+	−	−	−

BWG: Bland–White–Garland anomaly; MI: mitral insufficiency; PS: pulmonary stenosis; NA: not available/not assessed.

## Data Availability

The data presented in this study are available on request from the corresponding author.

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
