# Peer review of "Helsmoortel–Van der Aa Syndrome—Cardiothoracic and Ectodermal Manifestations in Two Patients as Further Support of a Previous Observation on Phenotypic Overlap with RASopathies"

_genes, 2022, doi:10.3390/genes13122367_

Round 1

Reviewer 1 Report

The authors presented two patients with ADNP mutations and listed all the available clinical features of the patients. These two patients have nonsense or frameshift mutation that causes ADNP protein loss of function.

This reviewer wonder if the authors have any evidence about the relationship between a specific mutation and certain clinical features. For example, if there are any cases with the same mutation that show similar symptoms. 

The authors also brought up the ketamine trial for patients/parents to consider. Maybe the authors could include more information regarding the ADNP mutation included in the trial for the readers to know if the treatment was only specific to certain mutation and if there is any information regarding the improvement of cardiothoracic and ectodermal manifestations, which would be relevant to this manuscript.

Author Response

Dear Reviewer,

Thank you for reviewing our manuscript, thank you for your comments and suggestions – we believe that our manuscript benefited from the applied changes based on your recommendations.

Other than implementing the recommended changes, we refined the interpretation of one important clinical feature: when evaluating the pectus carinatum in Patient 1 as a feature overlapping with RASopathies, a careful approach considering the clinical history with cardiac surgery and sternotomy as a contibuting factor was taken. We searched and found evidence that bilateral pectus carinatum arises from overgrowth of the costal cartilages and undergrowth of the bony ribs that supported our thinking that pectus carinatum is rather a malformation than a deformation in our patient (Park et al. The etiology of pectus carinatum involves overgrowth of costal cartilage and undergrowth of ribs. Journal of Pediatric Surgery, 49:8; 1252-1258).

We completed the references and text accordingly.

Below please see our answers to Your comments and suggestions.

Point 1: The authors presented two patients with ADNP mutations and listed all the available clinical features of the patients. These two patients have nonsense or frameshift mutation that causes ADNP protein loss of function. This reviewer wonder if the authors have any evidence about the relationship between a specific mutation and certain clinical features. For example, if there are any cases with the same mutation that show similar symptoms.

Response 1: Yes; in their paper Van Dijck et al. (Clinical Presentation of a Complex Neurodevelopmental Disorder Caused by Mutations in ADNP, Biol Psychiatry. 2019 Feb 15;85(4):287-297.) concluded that there are some recurrent hot-spot mutations affecting the 719, 730, and 831-832 amino acid positions (p.Tyr719*, p.Leu831Ilefs*82/p.Asn832Lysfs*81, and p.Arg730* mutations). Out of these, patients having the p.Tyr719* mutation walked later and had a higher pain threshold than the individuals with other mutations. Patients with mutations in the C terminal of the nuclear localization signal domain (NLS in their molecular figure; presumably patients with pathogenic variants in the 730, 736, 738 amino acid positions) more often had ptosis or oral movement problems than individuals with mutation elsewhere in the gene. They did not find genotype-phenotype correlation for age-, gender- or IQ-level, albeit ADNP was found to be differentially expressed in the male and female brain of animal subjects (Kleiman G, Barnea A, Gozes I. ADNP: A major autism mutated gene is differentially distributed (age and gender) in the songbird brain. Peptides. 2015 Oct;72:75-9.). Higher levels of ADNP mRNA were specifically found in young male compared to the female cerebrum, while aging caused a significant 2 and 3-fold decrease in the female and male cerebrum, respectively.

Our Patient 1 has the same mutation as Patients 28 and 39 in the Van Dijck 2019. paper: p.(Val180Glyfs*17), and there is striking similarity between our Patient 1 and their Patient 39: the upward gaze, facial gestalt are almost identical, and as far as we can judge that patient also has thin – but not sparse – hair. Our Patient 2 has the same mutation (only not C>A and not C>G, but the amino acid consequence is the same, p.Ser738*) as their patient 71. Individual phenotypic description for these patients is not availabe in their supplementary table. We did not find facial photograph of an individual with the p.Ser738* variant either in the Van Dijk et al, nor in the Helsmoortel et al., 2014. cohort (A SWI/SNF-related autism syndrome caused by de novo mutations in ADNP. Nat Genet, 46(4):380-4.).

Point 2: The authors also brought up the ketamine trial for patients/parents to consider. Maybe the authors could include more information regarding the ADNP mutation included in the trial for the readers to know if the treatment was only specific to certain mutation and if there is any information regarding the improvement of cardiothoracic and ectodermal manifestations, which would be relevant to this manuscript.

Response 2: We looked for the answer to this question on both the clinical trial’s website (clinicaltrials.gov) and in available literature (Kolevzon et al. An open-label study evaluating the safety, behavioral, and electrophysiological outcomes of low-dose ketamine in children with ADNP syndrome. Human Genetics and Genomics Advances 3, 100138, October 13, 2022) and we found no selection criteria for the study based on genotype. 6/10 patients had a frameshift variant, 4/10 had a nonsense variant. We included this in the paper. No RNA sequencing was performed to measure changes in ADNP expression, only clinical changes were recorded.

Also, please see our explanation for the pectus carinatum in paragraph 2 of the cover letter and in the text. The corrected manuscript is attached below

Debrecen, Hungary, 10 December 2022.

Yours sincerely,

Tímea Margit Szabó MD

Reviewer 2 Report

 The authors describe two cases with Helsmoortel-van der Aa syndrome with previously unreported clinical manifestations contributing to the understanding of this entity. Table 1 provides a useful comparison. In my opinion, only minor fixes are required. 

Keep “Helsmoortel-van der Aa” uniform and consistent in the text some places (Abstract, line 211) goes as “Helsmortel-van der AA syndrome

Line 262-263; Helsmortel-Van der Aa > Helsmoortel-Van der Aa

Line 103, and Table 1; ADNP should be in italics as it indicates the gene

Line 107; born at term

Line 108-109; give percentiles in parenthesis

e.g.

normal birth parameters; weight: 3150 gram (50th pc); length: 50 cm (50th pc), head circumference: 33 cm (25th-50th pc).

Line 110; prompted cardiological examination > prompted echocardiography

Line 115; “Angiography and coronary CT “? are these two tests or, do you mean computerized tomography (CT) coronary angiogram?

Line 168-169; lowing birth parameters: weight; 2480 gram (10th-25th pc), length; 44 cm (1.5 cm < 3rd  pc), head circumference; 33.5 cm (50th pc).

LINE 168; 2480 gramm > 2480 gram

Line 169; 33,5 cm > 33.5 cm

Line 181-182; give percentiles in parenthesis

e.g.

somatic parameters were normal; weight ; 14.5 kg (50th -75th pc); length ; 90.5 cm (10th-25th pc); head circumference ; 49.5 cm (50th pc).

If you have assessed the severity of ASD with CARS indicate the score.

Add Bland-Garland-White syndrome to Table 1

What are the ectodermal findings you have observed in these patients? discuss this..

Author Response

Dear Reviewer,

Thank you for reviewing our manuscript, thank you for your comments and suggestions – we believe that our manuscript benefited from the applied changes based on your recommendations.

Other than implementing the recommended changes, we refined the interpretation of one important clinical feature: when evaluating the pectus carinatum in Patient 1 as a feature overlapping with RASopathies, a careful approach considering the clinical history with cardiac surgery and sternotomy as a contibuting factor was taken. We searched and found evidence that bilateral pectus carinatum arises from overgrowth of the costal cartilages and undergrowth of the bony ribs that supported our thinking that pectus carinatum is rather a malformation than a deformation in our patient (Park et al. The etiology of pectus carinatum involves overgrowth of costal cartilage and undergrowth of ribs. Journal of Pediatric Surgery, 49:8; 1252-1258).

We completed the references and text accordingly.

Below please see our answers to Your comments and suggestions.

Point 1: Keep “Helsmoortel-van der Aa” uniform and consistent in the text some places (Abstract, line 211) goes as “Helsmortel-van der AA syndrome”

Line 262-263; Helsmortel-Van der Aa > Helsmoortel-Van der Aa

Response 1: We changed these typos to „Helsmoortel-Van der Aa” according to the official spelling in OMIM, and remained consistent with it throughout the paper.

Point 2: Line 107; born at term

Response 2: Corrected to „born at term”. It is now in line 112.

Point 3: Line 103, and Table 1; ADNP should be in italics as it indicates the gene

Response 3: Both corrected to italics.

Point 4: Line 108-109; give percentiles in parenthesis, e.g. normal birth parameters; weight: 3150 gram (50th pc); length: 50 cm (50th pc), head circumference: 33 cm (25th-50th pc).

Response 4: Corrected, percentiles are now in parenthesis.

Point 5: Line 110; prompted cardiological examination > prompted echocardiography

Response 5: Corrected; thank you for rephrasing.

Point 6: Line 115; “Angiography and coronary CT “? are these two tests or, do you mean computerized tomography (CT) coronary angiogram?

Response 6: From the patients’ medical records we understood these were two tests. One was performed via catheterization of the coronary arteries (under X-ray) with contrast material and the other was a coronary artery CT – the use of contrast material here was not disclosed specifically – we assumed it was a CT angiogram.

Point 7: Line 168-169; birth parameters: weight; 2480 gram (10th-25th pc), length; 44 cm (1.5 cm < 3rd  pc), head circumference; 33.5 cm (50th pc).

Response 7: Corrected according to your suggestion.

Point 8: LINE 168; 2480 gramm > 2480 gram, Line 169; 33,5 cm > 33.5 cm, Line 181-182; give percentiles in parenthesis, e.g.: somatic parameters were normal; weight ; 14.5 kg (50th -75th pc); length ; 90.5 cm (10th-25th pc); head circumference ; 49.5 cm (50th pc).

Response 8.: Corrected according to request. Lines 181-182 now and lines 195-196 in the current form of the MS now, respectively.

Point 9: If you have assessed the severity of ASD with CARS indicate the score.

Response 9: No, unfortunately we did not. This is the reason why we phrased „…suggesting autism spectrum disorder”. Symptoms regarding behaviour, communication, tolerance of social situations, speech were pointing toward autism, but no scoring system was used to assess severity. A lesson learned from your question that autism should be evaluated by objective means.

Point 10: Add Bland-Garland-White syndrome to Table 1

Response 10: This feature can be found in both tables as an abbreviation because of the lack of long cells and space. Full terms for the abbreviations are displayed under the tables.

Cardiac defect, unspecified

8.7%

+ (BWG, MI)

+ (PS)

Point 11: What are the ectodermal findings you have observed in these patients?

Response 11: The discussion is now completed accordingly. (sparse, thin, hair, thin, brittle nails, hypotrichosis of the eyebrows, early primary dentition – latter is not a RASopathy feature but an ectodermal feature.

Also, please see our explanation for the pectus carinatum in paragraph 2 of the cover letter and in the text.

Debrecen, Hungary, 10 December 2022.

Yours sincerely,

Tímea Margit Szabó MD
